# Multiplex real-time PCR using SYBR Green: Unspecific intercalating dye to detect antimicrobial resistance genes of *Streptococcus pneumoniae* in cerebrospinal fluid

**Mariana Brena Souza, Maria Cecília Cergole-Novella, Delma Aparecida Molinari, Daniela Rodrigues Colpas, Andréia Moreira dos Santos Carmo, Vilma dos Santos Menezes Gaiotto Daros, Ivana Barros de Campos** *

Santo André Regional Center, Adolfo Lutz Institute, Santo André, São Paulo, Brazil

* ivanacamp@gmail.com, ivana.campos@ial.sp.gov.br

**Data Availability Statement:** All relevant data are within the paper.

## Abstract

Meningitis caused by *Streptococcus pneumoniae* is still a disease of great impact on Public health, which requires immediate diagnosis and treatment. However, the culture of clinical specimens is often negative and antibiotic susceptibility testing (AST) must be performed with isolated strains. Multiplex real-time polymerase chain reaction (qPCR) has high sensitivity and specificity, produces faster results to identify the pathogen, and it can also be an important tool to identify resistance antibiotic genes earlier than AST, especially in the absence of an isolated strain. This study developed a multiplex qPCR assay, using SYBR Green as a nonspecific dye, to detect antibiotic resistance genes to predict pneumococcal susceptibility/resistance in cerebrospinal fluid (CSF) samples from meningitis patients. From 2017 to 2020, CSF samples were cultured and analyzed by qPCR to detect the main three bacteria causing meningitis. Isolated and reference strains were applied in SYBR Green qPCR multiplex to detect *pbp2b*, *ermB*, and *mef* genes, and the results were compared with the AST. Pneumococcal-positive CSF samples (*lytA*-positive gene) without isolated strains were also tested to evaluate the antimicrobial susceptibility profile in the region from 2014 to 2020. From the received 873 CSF samples; 263 were cultivated, 149 were *lytA*-positive in the qPCR, and 25 produced viable isolated pneumococci strains, which were evaluated by AST. Melting temperature for each gene and the acceptance criteria were determined (*pbp2b*: 78.24–79.86; *ermB*: 80.88–82.56; *mef*: 74.85–76.34 ºC). A total of 48/51 strains presented a genetic profile in agreement with the AST results. Resistant strains to erythromycin and clindamycin were *ermB*-positive, and two were also *mef*-positive, indicating both resistance mechanisms were present. In the retrospective study of the genetic profile of resistance, 82 *lytA*-positive CSF samples plus 4 strains were applied in the SYBR Green qPCR multiplex: 51% of samples presented the wild genotype (*pbp2b* positive and *ermB*/*mef* negative); 15% were negative for all the three evaluated, indicating pneumococci resistant to penicillin; and 17% represented the multidrug-resistant pneumococci (*pbp2b*

**Funding:** This work was supported by The São Paulo Research Foundation, FAPESP grant 2017/03022-6 (IBC) and 2018/22718-4 (MBS). The funders had no role in study design, data collection and analysis, decision to publish, or preparation of the manuscript. There was no additional external funding received for this study.

**Competing interests:** The authors have declared that no competing interests exist.

negative and *ermB* positive or *pbp2b* negative and *ermB* and *mef* positive). Therefore, SYBR Green qPCR multiplex proved to be a reliable tool to identify resistance genes in *S. pneumoniae* and would be less expensive than multiplex qPCR using specific probes. This could be easily introduced into the routine of diagnostic laboratories and provide a strong presumption of pneumococcal resistance, especially in the absence of isolated strains.

## Introduction

Meningitis is an inflammation of the defensive membranes covering the brain and spinal cord. Bacterial meningitis is a disease of great impact on Public health and can be caused mainly by *Streptococcus pneumoniae*. Meningitis due to *S. pneumoniae* occurs worldwide most commonly in infants and elderly, with an estimated incidence rate of 17 cases per 100,000 population in children less than five years of age [1]. From 2007 to 2016, in Brazil, 207,494 cases of meningitis were identified; of these, 102,249 were bacterial meningitis, of which 10,678 were related to *S. pneumoniae* (10.4%), and with a higher incidence in the southeastern region of Brazil (50.1%) [2]. Bacterial meningitis requires immediate diagnosis and treatment, due to the potential morbidity and high lethality. However, the cultures of clinical specimens and antibiotic susceptibility testing (AST) are slow and often negative. Modern techniques, such as the multiplex real-time polymerase chain reaction (qPCR) technique, have been applied to identify the main species causing meningitis and it was introduced by Corless et al. [3]. qPCR has high sensitivity and specificity, and produces faster results because it does not require the previous growth of the microorganism. Multiplex qPCR of the cerebrospinal fluid (CSF) proved to be a valuable method for improving the rapidity and accuracy of a diagnosis of bacterial meningitis, even in cases with CSF culture-negative results [4,5].

Besides the precise diagnostic, it is important that the information about the antibiotic susceptibility profile of the strain that is causing meningitis. This can lead to a good prognosis for the patient, as well as expand epidemiological information for Public health surveillance. For this purpose, nowadays, it is also necessary the culture of the clinical specimens to perform AST and the evaluation of strain phenotype. Notwithstanding, the issue of often culture-negative results is the same. It is known that resistance to beta-lactam antibiotics in pneumococci is due to alterations in the amino acid sequence of the PBP2x, PBP2b, and PBP1a penicillin-binding proteins [6] driven by selection with the excessive use of antibiotics. Real-time PCR is highly accurate in diagnosing meningitis caused by *S. pneumoniae* [4,5,7] and it also can be applied as a tool to detect antimicrobial resistance genes faster than the identification of phenotype by AST, especially in culture-negative samples.

Therefore, this study aims to develop a multiplex qPCR assay, using SYBR-Green as a nonspecific dye, to detect genes that confer resistance to *S. pneumoniae* in CSF samples from patients with pneumococcal bacterial meningitis. This technique would be less expensive than multiplex qPCR using specific probes and could detect resistance genes even in the absence of isolated strains.

## Material and methods

### Human samples

Cerebrospinal fluid (CSF) samples were collected from suspected cases of bacterial meningitis from 2014 to 2020 in health units in six municipalities (Diadema, Mauá, Santo André, São

Bernardo do Campo, São Caetano do Sul, Ribeirão Pires) of the State of São Paulo, southeastern Brazil. The CSF was delivered to the Adolfo Lutz Institute, Regional Laboratory of Santo André for epidemiological surveillance, as part of the diagnostic routine of our laboratory. In the period of 2017 to 2020, the raw samples were also inoculated on a brain and heart infusion (BHI) agar plate with 5% horse blood chocolate and incubated for 24 to 48 hours at 37˚C, using an anaerobic atmosphere [8] in the attempt to isolate strains causing meningitis. Isolated strains were identified by microbiological classic techniques. The genetic materials from all samples were extracted by silica column kits, according to the manufacturer's instructions, or by heating, as previously standardized [9,10]. After extraction, all samples were submitted to the multiplex qPCR developed by Adolfo Lutz Institute of São Paulo [4 and modified by 11], to identify the main three bacteria causing meningitis: *N. meningitidis* (*ctrA* gene), *S. pneumoniae* (*lytA* gene) and *H. influenzae* (*hpd* gene), as part of the diagnostic routine of our laboratory.

The Institutional Ethical Committee has approved this work by CAAE nº 80295817.7.0000.0059 and waived the informed consent from patients for this study and forthcoming publication since clinical specimens reported here were collected for diagnostical purposes and data were analyzed anonymously.

## SYBR Green qPCR multiplex

Multiplex qPCR technique was performed using the non-specific intercalating dye SYBR Green to detect different genes found in pneumococcus associated with resistance to antibiotics in the same reaction: *pbp2b* (penicillin-binding protein 2b gene) indicating a wild type strain and thus its sensitivity to beta-lactams; *ermB* (erythromycin ribosomal methylase B gene) which confers resistance to macrolides and lincosamides; and *mef* (macrolide efflux gene) which confers resistance to macrolides. Primers forward and reverse, except probes, from qPCR protocol described by the Centers for Disease Control and Prevention (CDC) [12] were applied, with one modification: primer *mef* reverse was GGTGTGAAAAGCTGTTCCAA. This alteration was necessary to obtain three fragments with different melting temperatures ($T_m$). Different primer concentrations were tested to combine the six primers in the same reaction. It was applied PowerUp™ SYBR™ Green Master Mix (Thermo Fisher Scientific, USA), according to the manufacturer's instructions, using the standard cycling mode for primers with $T_m < 60$˚C, followed by the dissociation curve. It was used the LightCycler® 480 II (Roche, Switzerland) with the following configuration of 'detection formats': excitation filter 483 nm, emission filter 533 nm, melt factor 1.2, quant factor 4, and max integration time 1 sec. The thermal cycling was performed at 50˚C for 2 minutes for UDG activation; 95˚C for 2 minutes for Dual-Lock™ DNA polymerase; and then 40 cycles of 95˚C for 15 seconds, 58˚C for 15 seconds, and 72˚C for 1 minute. Dissociation curve conditions were in three steps: (1) 95˚C for 15 seconds with 1.6˚C/second of ramp rate; (2) 60˚C for 1 minute with 1.6˚C/second of ramp rate and (3) 95˚C with 0.14˚C/second of ramp rate. The acquisition mode was continuous in this third step, with four acquisitions per ˚C. The qPCR products were differentiated by the $T_m$ obtained in the dissociation curve. Different samples were tested with the protocol standardized in this study: DNA from pneumococcus strain purified using silica column kits (for example PureLink™ Genomic DNA Mini Kit—Thermo Fisher Scientific, USA or QIAamp® DNA Mini Kit—Qiagen, Germany); and DNA from CSF samples, which were previously applied in the qPCR for bacterial meningitis, as described in the previous subsection and categorized as *lytA*-positive CSF samples, and were purified using silica column kits or extracted by heating (100 ˚C for 5 minutes) in a dry block.

## Analysis

Amplified fragments (amplicons) were evaluated *in silico* about their probable $T_m$. It was applied the uMELT Quartz—Melting Curve Predictions Software [13], set for 'Blake & Delcourt (1998)' thermodynamics to estimate the temperature where helicity is around 50%.

In the melting curve analysis, the $T_m$ was determined empirically for all genes in each sample. Firstly, all the $T_m$ data were collected and the mean of each replicate was defined when the SYBR Green qPCR multiplex condition standardized here was applied. Then, it was calculated the mean for each condition and the interquartile range (IQR 25–75). The normal distributions of the $T_m$ values for each gene were tested by the Kolmogorov-Smirnov test (KS normality test), D'Agostino & Pearson omnibus normality test, and Shapiro-Wilk normality test, to determine whether they were parametric or not. The results obtained with DNA purified from strains and DNA purified from CSF samples by silica column or heating were compared to the hypothetical value calculated by uMELT by one-sample t-test (parametric) or Wilcoxon signed-rank test (nonparametric). Then, the $T_m$ from all groups for each gene were compared to determine the statistical difference by one-way analysis of variance (ANOVA) or unpaired t-test. The analyses were performed in the GraphPad Prism version 5.0, and P-value < 0.05 was considered to represent statistical significance. Then, the acceptance criteria for a positive and a negative sample were generated. The $T_m$ range for acceptance of unknown samples was chosen by the lowest and the highest percentile for each gene. And the minimum fluorescence was empirically observed.

## Comparison of techniques

SYBR Green qPCR multiplex results using pneumococcus DNA from the strains isolated in this study and reference strains, which were obtained from the Center for Interdisciplinary Procedures of Microorganisms Collection of the Adolfo Lutz Institute of São Paulo, affiliated to the World Federation Culture Collections (WFCC), were compared with the AST results performed by Adolfo Lutz Institute of São Paulo, according to CLSI protocols [14]. Briefly, strains were tested by disk diffusion methodology or minimum inhibitory concentration (MIC), depending on the tested antibiotic. CSF samples that were culture-positive for pneumococcus were also submitted to the SYBR Green qPCR multiplex to compare with the DNA purified from the strain. Moreover, *lytA*-positive culture-negative CSF samples and *lytA*-positive not cultivated CSF samples were also tested by SYBR Green qPCR multiplex to evaluate the antimicrobial susceptibility profile of pneumococcus strains that caused meningitis in the region from 2014 to 2020.

## Results

### Characterization of analyzed samples

In the period from January/2014 to September/2020, 873 CSF samples from patients suspected of bacterial meningitis were received, and 149 were *lytA*-positive in the qPCR applied in the routine of the laboratory, which means they were positive for *Streptococcus pneumoniae*. From 2017 to 2020, 263 CSF samples were received with enough volume and therefore were cultivated, and 26 samples were culture-positive for pneumococci. Of these, 25 isolated *S. pneumoniae* strains were submitted to the AST, because one strain was not viable in further culture. Moreover, 26 pneumococcus reference strains, which had the antibiotic resistance profile previously characterized at the Adolfo Lutz Institute of São Paulo, were added in the analyses of this study to promote better statistic calculation, thus a total of 51 strains was applied in this study. From all 149 *lytA*-positive CSF samples, 82 (55.0%) had enough volume to proceed with

the SYBR Green qPCR multiplex for antimicrobial susceptibility genes identification. Twenty-one of 82 (25.6%) CSF samples had the antibiotic resistance profile known, due to the isolated strain. Therefore, four isolated strains did not have more CSF samples for further analysis and the results of these 21 CSF samples were compared with the results obtained from purified DNA of isolated strains. On the contrary, 61/82 had the antibiotic resistance profile unknown, which could not be compared to an isolated strain, but it was used for retrospective study in the region.

## Standardization of the SYBR Green qPCR multiplex

Firstly, to differentiate the amplicons that would be formed, the $T_m$ for each gene was evaluated *in silico*. Hence, it would be possible to create a triplex protocol. The calculated $T_m$ were 79.5 ˚C for *pbp2b*, 82 ˚C for *ermB*, and 77 ˚C for *mef* genes. It was observed that the prediction varied according to the software and the setup applied. Some online tools and the software applied here with other settings have calculated $T_m$ far outside from the experimentally observed. Then the assay was performed to check the real $T_m$ for each gene. Secondly, different annealing temperatures (54, 56, 58, and 60˚C) were tested using the same master mix. The best temperature observed for all tested samples was 58˚C. Also, after extensive tests, the final concentration of 0.3 μM was standardized to each primer when DNA purified from isolated strains was applied and 0.5 μM to each primer when DNA was obtained from CSF samples. These concentrations and annealing temperature were extremely important to avoid false positives and false negatives.

## Statistical analyses of the SYBR Green qPCR multiplex

After the assays were performed, the melting curves were checked and the variances of $T_m$ results were analyzed. For each gene, there were three evaluated groups: values of $T_m$ after qPCR using purified DNA from cultivated strain; from CSF sample using silica column; or heating. The $T_m$ values distributions were considered parametric for only three groups, as presented in Table 1. Additionally, each group mean was compared to the hypothetical $T_m$ calculated by uMELT Software. Most of them were statistically different from the hypothetical $T_m$ (Table 1). Furthermore, the groups for DNA purified by silica column or by heating from CSF samples were compared to check the possibility to unite as one group (CSF samples) per gene. It was applied unpaired t-test (Student's) and a significant difference was observed. For this reason, these results remained separated for further analyses. However, the *mef* group presented a small number of samples; consequently, the values of $T_m$ were unified in one group of CSF samples. The acceptance criteria for both types of samples (strain and CSF) were established: it was considered a positive sample when the SYBR Green qPCR multiplex presented an amplification curve that cross the threshold line, thus presenting the cycle threshold (CT) positive. Also, the dissociation curve must be analyzed to confer if the sample presented a $T_m$ peak within the acceptance $T_m$ range for any gene (Table 1). Moreover, this peak (-d/dT Fluorescence at 465–510) must have a height superior to one when DNA purified from strains was applied, and any height value when DNA purified from CSF sample was applied, since the $T_m$ was within of the suggested here.

Finally, the ANOVA test was applied, and the mean of $T_m$ for *pbp2b* and *ermB* genes groups were statistically different (P < 0.0001). Tukey's Multiple Comparison Test was performed to compare all pairs of means. Fig 1 presents the mean of $T_m$ for all replicates for each strain or CSF sample, and all genes separately. It also presents the mean of all observed values with the standard deviation. It was presented as well the Tukey test P-value. For *mef* gene, as the groups presented a small number of samples and the results were unified in one group of CSF samples,

**Table 1. Statistic analyses of melting temperature ($T_m$) values obtained in the dissociation curve after qPCR using DNA purified from cultivated strain and purified from cerebrospinal fluid (CSF) sample.**

| | *pbp2b* | | | *ermB* | | | *mef* | |
|---|---|---|---|---|---|---|---|---|
| | **Strain** | **CSF (silica)** | **CSF (heating)** | **Strain** | **CSF (silica)** | **CSF (heating)** | **Strain** | **CSF (silica and heating)** |
| Number of values | 39 | 11 | 8 | 11 | 4 | 3 | 3 | 3 |
| Minimum | 78.22 | 77.98 | 79.39 | 81.28 | 80.81 | 82.07 | 75.19 | 74.85 |
| Maximum | 79.03 | 78.83 | 79.90 | 81.51 | 81.19 | 82.56 | 75.23 | 76.34 |
| Mean ± SD | 78.74 ±0.13 | 78.42 ±0.23 | 79.69 ±0.21 | 81.36 ±0.07 | 81.06 ±0.17 | 82.27 ±0.26 | 75.21 ±0.02 | 75.38 ±0.83 |
| IQR 25–75 | 78.67–78.81 | 78.24–78.53 | 79.44–79.86 | 81.30–81.40 | 80.88–81.18 | 82.07–82.56 | 75.19–75.23 | 74.85–76.34 |
| Normality test | No | Passed | Passed | Passed | Too small [a] | Too small [a] | Too small [a] | Too small [a] |
| Coefficient of variation | 0.16% | 0.29% | 0.26% | 0.09% | 0.21% | 0.32% | 0.02% | 0.97% |
| One sample t test (P-value) | < 0.0001 | < 0.0001 | 0.0387 | < 0.0001 | 0.0016 | 0.2194* | < 0.0001 | 0.0779* |
| Acceptance criteria | 78.24–79.86 | | | 80.88–82.56 | | | 74.85–76.34 | |

Strain: DNA purified from isolated strains; CSF silica: DNA purified from CSF samples using silica column; CSF heating: DNA purified from CSF samples by heating; number of values: Number of different samples tested for each condition; Minimum: The smallest value of $T_m$ obtained; Maximum: The largest value of $T_m$ obtained; Mean ± SD: Mean and the standard deviation; IQR 25–75: The interquartile range of the 25th and 75th percentile; Normality test: Results obtained from AP test and SW test to check normal distribution. Passed implies P-value > 0.05, which means that the data are consistent with a Gaussian distribution; Coefficient of variation: The standard deviation divided by the mean; One sample t test: P-value obtained from comparison of the mean to the hypothetical value estimated by uMELT; Acceptance criteria: The acceptance $T_m$ range suggested in this study.

[a] considered normal distribution by the KS normality test.

* No significant difference to the hypothetical value was observed.

unpaired t-test was applied. The two groups were considered not statistically different, P = 0.7389 (Fig 1).

## Genotypes versus antimicrobial sensitivity profiles

Of 51 pneumococci strains, 39 were positive for the *pbp2b* gene, which means they have the wild genotype and thus should be susceptible to penicillin (Table 2). Also, 11 strains were

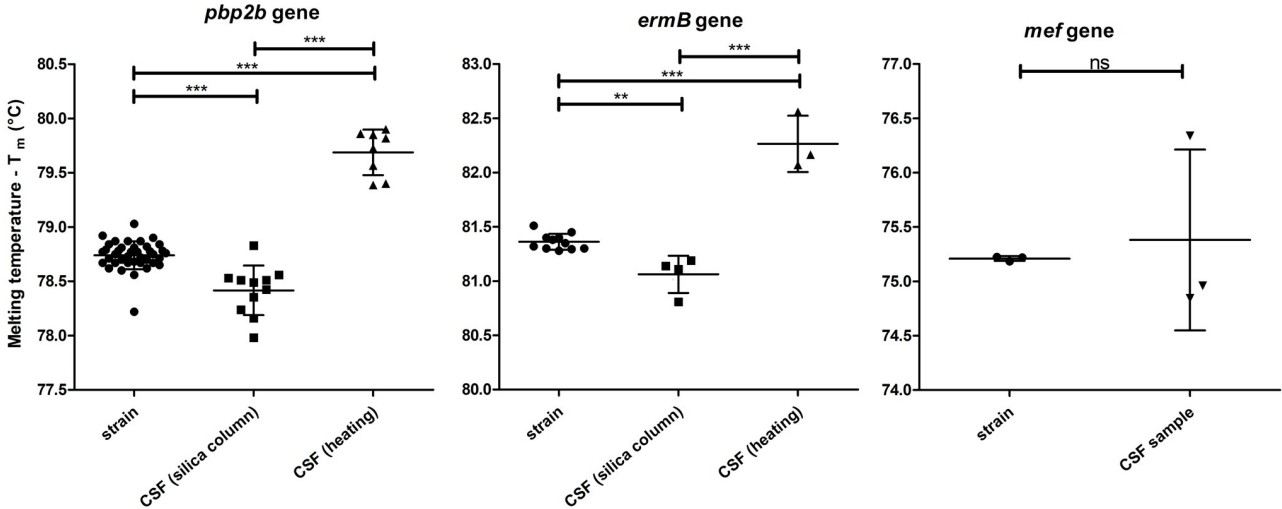

**Fig 1. Mean of the Melting Temperature (Tm) values for all replicates for each DNA extracted from strain or CSF sample by silica column or by heating.** Each column represents the data obtained for one gene which is identified at the top. Short horizontal bars indicate the mean; Vertical bars show the standard deviation; Long horizontal bars represent the significance of differences calculated by the Tukey's Multiple Comparison Test and shown by asterisks: **P < 0.01 and ***P < 0.001. For the *mef* gene, it was calculated by unpaired t-test: ns means not significant (P > 0.05).

**Table 2. Antimicrobial sensitivity profile and genotypes of clinical isolates using SYBR Green qPCR multiplex.**

| No. of isolates | Susceptibility profile | | | Presence of genes | | |
|---|---|---|---|---|---|---|
| | **Penicillin** | **Clindamycin** | **Erythromycin** | *pbp2b* | *ermB* | *mef* |
| 34 | Susceptible | Susceptible | Susceptible | 34[a] | - | - |
| 6 | Susceptible | Resistant | Resistant | 5[b] | 6 | - |
| 6 | Resistant | Susceptible | Susceptible | - | - | - |
| 4 | Resistant | Resistant | Resistant | - | 4 | 2* |
| 1 | Resistant | Susceptible | Resistant | - | 1[c] | 1* |
| 51 | | | | 39 | 11 | 3 |

qPCR with the discordant result:

[a], for strain 357, the *pbp2b* gene is not amplified according to the acceptance criteria, and it presents susceptibility to penicillin;

[b], for strain 404, the *pbp2b* gene is not amplified, but it is susceptible to penicillin;

[c], for strain 444, *ermB* gene is positive, but the strain is susceptible to clindamycin.

This strain is also erythromycin-resistant and has both genes (*ermB* and *mef*).

* Total of three strains presented both *ermB* and *mef* genes.

positive for the *ermB* gene, which means they are expected to be erythromycin and clindamycin resistant, and only 3 were positive for the *mef* gene, which means they are expected to be erythromycin resistant. A total of 48/51 strains presented a genetic profile in agreement with the AST results. Only three were discordant as presented in Table 2, called strains 357, 404, and 444. It was observed that the phenotype of 11 strains was resistant to penicillin (MIC ≥ 0.125 µg/mL) and all of them did not present the *pbp2b* gene in the SYBR Green qPCR multiplex standardized in this study; however, one from 40 strains susceptible to penicillin was always negative for *pbp2b* gene in the standardized protocol (strain 404), and another did not present amplified *pbp2b* gene according to the acceptance criteria (strain 357) (Table 2). Curiously, the CSF sample of which strain 357 was isolated presented the SYBR Green qPCR multiplex results in concordance with the AST result. All of the 10 strains resistant to erythromycin and clindamycin were *ermB*-positive. Two from 10 were also *mef*-positive, indicating both resistance mechanisms were present in these strains. On the other hand, one strain, called 444, also presented positive results for both *ermB* and *mef* genes, but it was susceptible to clindamycin and resistant to erythromycin. This was the third strain in disagreement with the AST results. For all the three strains in divergence, the AST was repeated to confirm the antibiotic resistance phenotype.

Fig 2 presents an example of melting peaks for each gene combination presented in the strains evaluated in this study, using the SYBR Green qPCR multiplex protocol developed here. Five different combinations of resistance genes were detected, as presented in Table 2, and shown graphically in Fig 2. Additionally, when analyzing CSF samples, 21 samples had enough material to be tested by the SYBR Green qPCR multiplex and 19 were compatible with the AST results applied to the isolated strains from these CSF samples. Thus, only two were discordant. One CSF sample did not present the same result as its discordant strain, as described in this subsection.

## Retrospective study of the genetic profile of resistance

With the established protocol in this study, the genetic profile of resistance genes presented in *lytA*-positive CSF samples from 2014 to 2020 was evaluated, which did not have pneumococci isolated strains, as a retrospective study of the region. Fig 3 presents the data from the SYBR Green qPCR multiplex performed with all 86 samples received in the period and which had

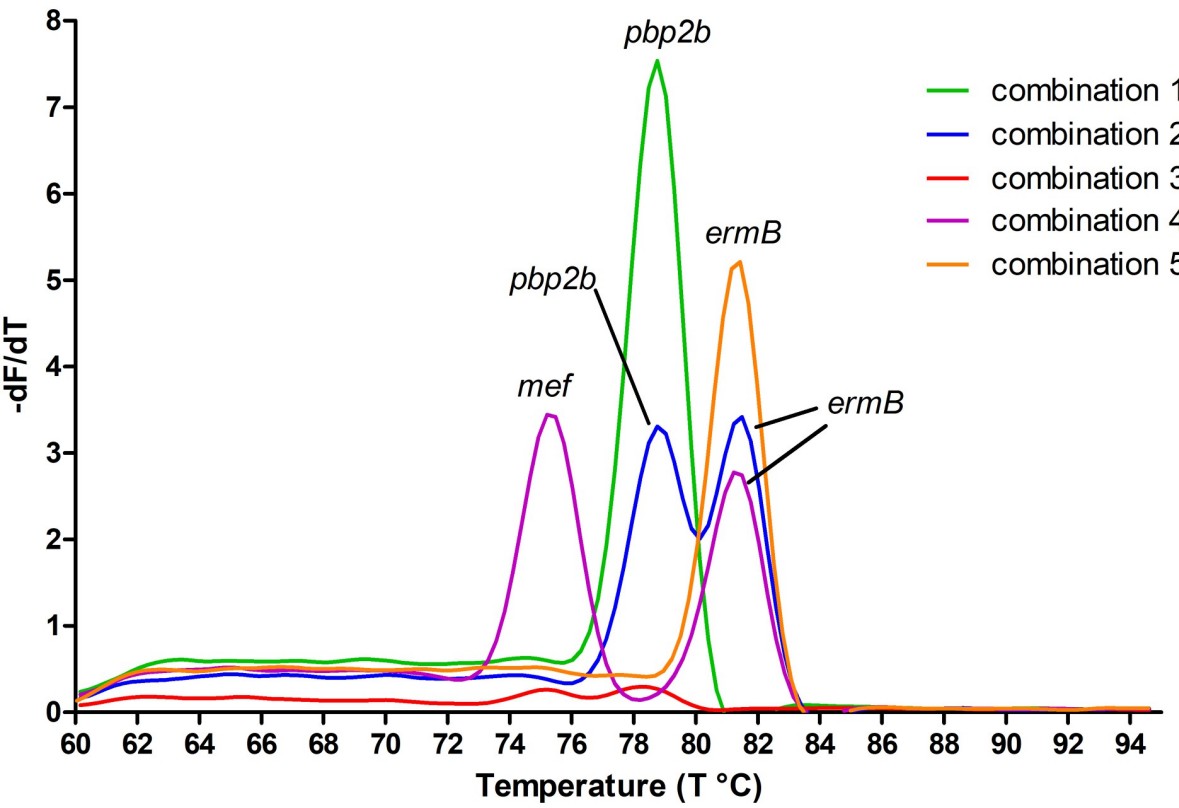

**Fig 2. Fluorescence melting peaks obtained by plotting the negative derivative of fluorescence over temperature (-dF/dT) versus temperature (T) in the SYBR Green channel (465–510 nm) for the SYBR Green qPCR multiplex.** Each gene combination observed in this study is represented here: Combination 1—susceptible to three classes of antibiotics, only *pbp2b* gene was positive; 2—susceptible to penicillin and resistant to clindamycin and erythromycin, *pbp2b* and *ermB* genes were positive; 3—resistant to penicillin and susceptible to clindamycin and erythromycin, all three genes were negative; 4—resistant to three classes of antibiotics, with multiples mechanisms of resistance, *pbp2b* was negative, *ermB* and *mef* genes were positive; 5—resistant to penicillin, clindamycin and erythromycin, *ermB* gene was positive. The peaks were identified with the name of the genes that they represent.

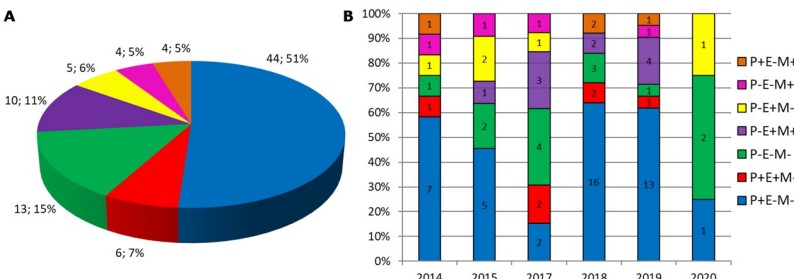

**Fig 3. Genetic profile of resistance genes detected in all strains or CSF samples received from 2014 to 2020: A— Distribution in percent of all 86 samples per genotype; B—Distribution in percent per year of all 86 samples.** The total of results per genotype is indicated inside of each bar; Genotypes: P+E-M- means *pbp2b* positive, *ermB* and *mef* negative; P+E+M- means *pbp2b* and *ermB* positive and *mef* negative; P-E-M- means *pbp2b*, *ermB* and *mef* negative; P-E+M+ means *pbp2b* negative, *ermB* and *mef* positive; P-E+M- means *pbp2b* and *mef* negative, and *ermB* positive; P-E-M+ means *pbp2b* and *ermB* negative, and *mef* positive; P+E-M+ means *pbp2b* and *mef* positive, and *ermB* negative.

enough material for the analyses per detected genotype. From all of the data, 61 were CSF samples with antibiotic resistance profile unknown due to the absence of the isolated strain; 21 were CSF samples with antibiotic resistance profile known as a result of the AST performed with the pneumococcus isolated strain; and 4 were also results from SYBR Green qPCR multiplex, but obtained from strains samples since there was no CSF sample for further analyses. Thus 82 CSF samples and 4 strains samples. It is possible to observe (Fig 3) that the genotype *pbp2b* positive and *ermB* and *mef* negative was the majority for the total of samples, and per year, except 2017 and 2020. This would be a wild genotype, with a pneumococcus strain sensitive to penicillin, macrolides, and lincosamides, and represents 51% of the samples. The second genotype observed (15%) was negative for all three evaluated genes, which would mean a pneumococcus strain resistant to penicillin, but sensitive to macrolides and lincosamides. It is worthy to note, that these samples which did not present any amplification, were confirmed as the presence of genetic material by a performance of the single qPCR for the *lytA* gene also using SYBR Green, in order to check DNA's integrity after years of storage. Multidrug-resistant (MDR) pneumococci strains, defined as resistant to at least one agent in three or more antimicrobial classes [15] may be here represented by the genotypes: *pbp2b* negative and *ermB* positive; *pbp2b* negative and *ermB* and *mef* positive, which would mean strains resistant to penicillin, macrolides, and lincosamides. Both groups represented a total of 15 samples, 17% of all.

## Discussion

Multiplex qPCR technique was performed in this study using intercalating dye SYBR Green. Since this dye is nonspecific for DNA products, any amplification will lead to an increased fluorescent readout. Therefore, besides cycle threshold (CT) evaluation, melting temperature ($T_m$) must be also confirmed if the target gene was amplified. For this reason, firstly, it was evaluated the hypothetical $T_m$ for each gene. As observed in other studies, which compared different melting temperature calculation methods [16–18], diverse outputs were attained here. However, the uMELT Software [13] proved to be the most reliable tool. Moreover, when the tests were performed, it was noted that there are differences in the $T_m$ according to the material applied. When the groups were compared, different sources of the material, DNA purified from CSF samples or from strains recovered from the same CSF sample, presented $T_m$ slightly dissimilar which was statistically significant. This difference indicates that the experimental $T_m$ obtained in the dissociation curve after qPCR is very sensitive to variation with the source of the genetic material. This is the first time, as far as the authors are aware, that the same genes from different sources present different $T_m$ empirically. Thus, an interval must be adopted to be considered positive according to the origin of the genetic material. Interestedly, despite the $T_m$ being considered different between groups, when analyzed in agarose gel, it was not observed difference in the size (bases pairs—bp). However, it is known that ordinary agarose gel has a poor resolution and is not able to distinguish few bp as by polyacrylamide gels or sieving agarose gel, which can differentiate about 10 bp or by capillary electrophoresis [19] or else by conformation sensitive gel electrophoresis, which can distinguish single-base mutation [20].

With the information obtained about $T_m$, it was elaborated the criteria of acceptance for the SYBR Green qPCR multiplex with the three genes studied here. These criteria were suggested when using the same parameters: master mix, real-time PCR equipment, and software of analysis, to avoid false-positive results. Genetic material from strains seemed to be easier to be amplified, producing higher fluorescence, whereas DNA from CSF samples always presented lower peaks of fluorescence, probably because they have less bacterial material. Therefore, any

fluorescence for CSF samples was accepted, since the sample presented CT and an expected $T_m$. And for DNA purified from strain, it is suggested that the fluorescence peak must be higher than one. These criteria for acceptance were designed for the platform applied in this study (Roche Lightcycler 480 II), which is widely used in laboratories around the world. Thus, all setting details were exposed here to guarantee the reproducibility of the assay. However, the portability of the assay to other platforms is possible with some adjustments. For example, most of the real-time PCR equipment is automatic and does not have specific settings to configure, only choosing the SYBR Green detection. The statistical difference observed for Tm using different sources of DNA (strain, CSF from column, and heating) was not expected, but it will happen independently of the PCR platform because it is strongly believed that the reason for this observation was the sample quality: the presence of contaminants and human DNA. A necessary adaptation that should be considered is the fluorescence of the Tm peak for positive and negative samples. Different platforms present different fluorescence units. For example, the Roche system presents -d/dT Fluorescence at 465–510 nm, and the Tm peak must have a height superior to one for strain and any height for CSF sample; whereas QuantStudio™ 5 Real-Time PCR System (Applied Biosystems™, Thermo Fisher Scientific) presents different fluorescence unit in the melt curve plot (derivative -Rn), and the order of magnitude is in the thousands. Moreover, despite the required adaptation of this methodology, it is still worthy due to the low cost of this assay. The SYBR Green qPCR multiplex is supposed to cost US$ 15–20 as observed in the literature [21,22]. Here, it was estimated < 1 dollar considering only master mix and primers; whereas qPCR multiplex using three different probes, it was estimated the cost of US$ 8–15, considering commercial kits with master mix, primers, and in this case probes, available for molecular diagnostic of other infectious diseases. The time to run this in-house assay is longer than commercial PCR kits; however, its cost is considerably lower.

When comparing the SYBR Green qPCR multiplex with the antibiotic susceptibility testing applied to the strain, it was verified that only three were discordant. Two of them were negative to the *pbp2b* gene, but the strains were susceptible to penicillin. Probably, these findings were observed due to point mutations where the primers are annealed, preventing the annealing and gene amplification, but not preventing the production of the penicillin-binding protein making them sensitive to the penicillin in the AST. Similarly, negative detection of the *pbp2b* gene in susceptible strains by qPCR using specific probes was observed in another study [12]. In addition, another study analyzed the *pbp* gene by PCR–RFLP and found 24 different composite pattern profiles for three resistance genes: *pbp1a*, *pbp2b*, and *pbp2x*. For penicillin-susceptible strains, they found only one profile for *pbp1a* and *pbp2x* genes while *pbp2b* presented three different profiles. They mentioned it is not clear how some penicillin-susceptible *S. pneumoniae* isolates can harbor mutations in *pbp*2b without becoming non-susceptible to the penicillin [23]. Additionally, it was observed here that the MIC values for penicillin have corresponded to the presence or absence of the *pbp2b* gene: high fluorescence for the expected $T_m$, thus considered positive, for MICs ≤ 0.06 μg/mL; and no fluorescence for the expected $T_m$, consequently negative for MICs > 0.125 μg/mL). Except for the two strains mentioned above. Resistant strains with low MIC for penicillin (= 0.125 μg/mL) sometimes presented a peak for the expected $T_m$, however, with low fluorescence and it did not present CT.

The third strain in disagreement with the AST presented both *ermB* and *mef* genes, but it was susceptible to clindamycin and resistant to erythromycin. Therefore, it would be expected to be negative for the *ermB* gene and positive for the *mef* gene. It was hypothesized that this strain has the *ermB* gene, but due to some internal mutations or a deletion/addition that can produce an out-of-frame reading gene, the protein is not in the right conformation, and therefore this resistance mechanism does not work. To the best of our knowledge, there is no documentation about a strain double-positive (*ermB* and *mef* genes) with a phenotype

clindamycin-susceptible. It was observed erythromycin susceptibility and clindamycin resistance profile in 3 strains of *Streptococcus agalactiae* portraying *erm* and *lnuB* genes by Moroi et al. [24], and they suggested a loss-of-function of ErmB protein, which could be what happened with our strain, and it keeps the resistance to erythromycin due to the *mef* gene. Although macrolides and lincosamides are not used to treat meningitis, the genetic characterization of antimicrobial resistance of pneumococcal isolates, which can be performed by different molecular marker methods with the isolated strain, is crucial to monitoring the epidemiology of the pneumococcal disease. The methodology developed here showed to be a reliable tool, especially when there is no pneumococcus growth, which contributes to the pneumococcal surveillance.

After standardization of the SYBR Green qPCR multiplex, it was applied to all *lytA*-positive CSF with enough volume, which could produce a profile of antibiotic resistance of pneumococci strains that caused meningitis in the region from 2014 to 2020. It was possible to observe the prevalence of the wild-type strain (51%), which means strains susceptible to all antibiotics corresponding to the evaluated genes. Furthermore, MDR pneumococci had great representativeness, a total of 17% of all samples, with the genotype *pbp2b* negative and *ermB* positive or *pbp2b* negative and *mef*, *ermB* genes positive. However, it indicates that MDR pneumococcal invasive strains were less common in the region of this study, as observed in other works. For example, 33% of MDR strains were identified by Sharew et al. [15]; 56.3% were identified by Ma et al. [25]; and the nationwide surveillance following 10-valent pneumococcal conjugate vaccine introduction in Brazil detected 25% multidrug-resistance in 2017–2019 [26]. All these works performed MIC test for a great variability of antibiotics as the AST, which could explain the higher percents. Another finding was that 10 samples (11%) presented both *ermB* and *mef* genes, which included CSF with unknown AST and isolated strains, 70% of these were 19A serotype. In the 2000s, reports about pneumococci with both resistance genes were occasional, but the concurrence of the genes increased considerably worldwide [27], indicating the expansion of multi-resistant clones [28]. The amount of double-positive pneumococcus 19A serotype represented 8% of the total samples analyzed here. In Brazil, it was observed an increase of 7% to 16.4% in the incidence of invasive pneumococcal disease caused by serotype 19A, after the 10-valent pneumococcal conjugate vaccine introduction, especially in 2016–2017 [29]. The molecular detection of both *ermB* and *mef* genes in pneumococcal invasive strains was higher in comparison with other studies in Brazil (3 in 159 strains– 1.9%) during 2010–2012 [30] and 7% during 2012–2013 [31]. Probably, because these studies evaluated the genes' presence in isolated strains. Here, we evaluated CSF samples and isolated strains, which increased the chances of detection. And lastly, it should be mentioned the decrease in the number of samples in 2020 occurred probably due to the COVID-19 pandemic. It was supposed that the use of masks and social distance contributed to the drop in meningitis cases.

## Conclusions

Despite the differences observed in $T_m$ when applying the same gene from different sources, the SYBR Green qPCR multiplex proved to be a reliable and inexpensive tool to identify resistance genes in *Streptococcus pneumoniae*. This could be easily introduced into the routine of diagnostic laboratories and provide a strong presumption of pneumococcal resistance, even in the absence of isolated strain, which would improve and strengthen the surveillance of pneumococcal diseases.

## Acknowledgments

We thank the Adolfo Lutz Institute, Regional Laboratory of Santo André crew for technical support in processing the clinical samples and other routine activities, especially to Patricia de

Lima Vicente, Valeria dos Santos Candido, Gabriela de Carvalho Gonçalves, Carmelita Selles de Souza, Maranice Cesário, Maria Clarice Pereira da Silva and Maria Julia de Lima Lopes. We also thank the crew from Adolfo Lutz Institute of São Paulo for providing reference strains and their antibiotic resistance profile, and for the performance of the AST with the isolated strains in this study, especially Samanta Cristine Grassi Almeida and Tania Sueli de Andrade.

## Author Contributions

**Conceptualization:** Ivana Barros de Campos.

**Data curation:** Mariana Brena Souza, Delma Aparecida Molinari, Daniela Rodrigues Colpas, Ivana Barros de Campos.

**Formal analysis:** Mariana Brena Souza, Ivana Barros de Campos.

**Funding acquisition:** Andréia Moreira dos Santos Carmo, Vilma dos Santos Menezes Gaiotto Daros, Ivana Barros de Campos.

**Investigation:** Mariana Brena Souza, Delma Aparecida Molinari, Daniela Rodrigues Colpas, Ivana Barros de Campos.

**Methodology:** Mariana Brena Souza, Delma Aparecida Molinari, Daniela Rodrigues Colpas, Ivana Barros de Campos.

**Project administration:** Ivana Barros de Campos.

**Resources:** Andréia Moreira dos Santos Carmo, Vilma dos Santos Menezes Gaiotto Daros, Ivana Barros de Campos.

**Supervision:** Ivana Barros de Campos.

**Visualization:** Ivana Barros de Campos.

**Writing – original draft:** Maria Cecília Cergole-Novella, Ivana Barros de Campos.

**Writing – review & editing:** Maria Cecília Cergole-Novella, Andréia Moreira dos Santos Carmo, Vilma dos Santos Menezes Gaiotto Daros, Ivana Barros de Campos.

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
