## [Decision Letter · Decision Letter 0]

22 Mar 2022

PONE-D-22-02044Multiplex real-time PCR using SYBR Green: unspecific intercalating dye to detect antimicrobial resistance genes of Streptococcus pneumoniae in cerebrospinal fluidPLOS ONE

Dear Dr. Campos,

Thank you for submitting your manuscript to PLOS ONE. After careful consideration, we feel that it has merit but does not fully meet PLOS ONE’s publication criteria as it currently stands. Therefore, we invite you to submit a revised version of the manuscript that addresses the points raised during the review process.

The manuscript has been assessed by two reviewers; the comments are available below.

The reviewers have raised a number of concerns about the clarity in presentation of the work and the data, they recommend revisions to improve the clarity in presentation and writing and to provide a fuller outline of the main results.

Please carefully revise the manuscript to address all the points raised by the reviewers

We look forward to receiving your revised manuscript.

Kind regards,

Jose Melo-Cristino, M.D., Ph.D.

Academic Editor

PLOS ONE

Journal Requirements:

(This work was supported by The São Paulo Research Foundation, FAPESP grant 2017/03022-6 (IBC) and 2018/22718-4 (MBS).

The funders had no role in study design, data collection and analysis, decision to publish, or preparation of the manuscript.)

3. We noted in your submission details that a portion of your manuscript may have been presented or published elsewhere. 

(PLOS Biology (PBIOLOGY-D-21-03354).

They suggested that I could transfer this manuscript to PLOS ONE, and I agreed)

Reviewers' comments:

Reviewer's Responses to Questions

**Comments to the Author**

1. Is the manuscript technically sound, and do the data support the conclusions?

Reviewer #1: Partly

Reviewer #2: Partly

2. Has the statistical analysis been performed appropriately and rigorously? 

Reviewer #1: Yes

Reviewer #2: Yes

3. Have the authors made all data underlying the findings in their manuscript fully available?

Reviewer #1: Yes

Reviewer #2: Yes

4. Is the manuscript presented in an intelligible fashion and written in standard English?

Reviewer #1: Yes

Reviewer #2: No

5. Review Comments to the Author

Reviewer #1: The authors present an adaptation of a previously proposed assay, but now based on melting curve analysis, for the identification of penicillin, macrolide and lincosamide resistant pneumococci in culture negative CSF specimens. The authors advance no justification for adapting the existing assay. The paper then presents a lengthy description of the statistical analyses performed and of the method development which can be greatly shortened and simplified. It is unclear what can be expected from 4/51 strains used to validate the assay since they do not seem to have been tested for their susceptibility to clindamycin or clindamycin and erythromycin. Sensitivity and specificity seem to be high (only 3 strains out of 47 had inconsistent results with phenotypic assay). A major undiscussed point is the portability of the assay to a different platform than the one used by the authors, which can limit its applicability. The authors then analyzed 86 samples received at their center between 2014-2020 (but only 61 were culture negative). The paper should be shortened to focus on the method and its appropriateness to determine pneumococcal susceptibility in CSF samples. Although overall the English is acceptable the paper would still benefit from a revision of English use.

Major points

1) The authors should clearly state the benefits of adapting the existing probe-based assay to a melting-curve based assay. If the motivation is economical, a paragraph stating the price differences should be presented and a convincing argument presented to support this.

2) The detailed description of the statistical tests employed should be shortened and only the relevant sections included. The main message of the paper is the method and its validation, and the paper should focus on this. The entire section on the choice of Tm prediction software (and the related sections in the discussion) should be removed. It is nonsensical to choose a prediction software and method based on its agreement with the empiric determined Tm, as is stated in lines 379-380.

3) The paper should clearly state what were the validation conditions used. If this was against the PCR on the DNA extracted from all the strains or, which would me much more interesting, against the DNA extracted from the CSF from which some of the strains were isolated. The discussion of results with different PCR conditions than the ones used (ex: lines 291-297) should be deleted.

4) The susceptibility of all isolates to all targeted antimicrobials must be known (table 2).

5) Given the variations in melting curves seen when DNA was prepared from different sources, what would be the robustness of the assay when run in different platforms? Although the Roche Lightcycler 480 II is quite a popular platform it is by no means the only one. A few sentences discussing the potential portability of the assay to other platforms would be important for a wider interest in the paper.

6) The purpose of the retrospective analysis and how it could impact on the clinical management of the patients or how the information could help in the surveillance of pneumococcal infections in this area must be clarified or this section removed. The fact that macrolides and lincosamides are not used in the treatment of pneumococcal meningitis raises the question of the usefulness of determining the potential susceptibility to these antimicrobials, a point that should be addressed.

7) The discussion should be shortened and focused on the main message of the paper without repetition of what was stated in the results section

Other points

8) Line 121. The authors changed the reverse primer amplifying the mef gene from the previously published assay. Why was this done? A sentence should be included to explain this option.

9) English use: line 116 “and thus it is sensitivity” should be “and thus its sensitivity”; line 164 “Comparison of technics” should be “Comparison of techniques”; line 181 “Characterization of applied samples” should be “Characterization of analyzed samples”; line 291 and elsewhere “acceptation criteria” should be “acceptance criteria”

10) Lines 222-224. The central limit theorem is not applicable to 39 data points which is what is available for the pbp2 gene (table 1). This sentence must be removed.

11) Table 1. The P values have comas instead of points separating the decimals. This table repeat the data presented in figure 1. I would suggest offering the figure as supplementary material.

Reviewer #2: The paper describes validation of a previously published Taqman multiplex real-time PCR assay for detecting 3 antimicrobial resistance genes, pbp2b, erm B and mef and instead using primers with SYBR green intercalating dye. The authors describe their approach for validation and then apply the assay to several culture-negative samples previously identified as lytA positive indicating presence of S. pneumoniae.

Overall, I found the paper quite difficult to read in its current format and could use some review for correct English sentence structure and grammar.

Specific comments

Line 21-22 and 59. Not necessary to have capital letters for Public Health?

Line 71. It is not clear what you mean by qPCR “provides more complete results than the standard techniques”?

Line 164. “technics” should be “techniques”

Line 206-208. I am curious why you would need to have 2 different primer concentrations when you use isolate vs CSF DNA? Can you explain and would you be concerned with performance of the assay with varying amounts of DNA in your sample? Did you do a serial dilution series to determine LOD?

Line 267-269 and Table 1. It Is not clear why mef has only 2 groups for comparison? Why was CSF silica and CSF heating combined? Was it due to sample size?

Line 292-297. Do you think that this approach using SYBR green without a probe is accurate enough given that you mention false positive and negative results with changes in temperature and primer concentrations. I did not see any data provided showing effects of sample concentration of DNA on the Tm and positive vs negative results. Can you explain in Fig 1 the large differences between Tm for CSF (silica) vs CSF (heating)? Is this related to DNA concentrations?

Line 387-388. You state “Thus, an interval must be adopted to be 388 considered positive according to the origin of the genetic material”. I have my doubts on how accurate this method would be if say for example you used a pleural fluid or blood sample? Would you be confident that this assay would accurately determine penicillin, erythromycin or clindamycin susceptibility in these other specimen types give the variability of the assay?

Lines 399-402. Again, here you indicate that differences such as changes in PCR instrumentation would affect results? I don’t see how this method could be useful to a larger community? Perhaps validated in your own lab for testing but seems problematic in that every lab would need to revalidate the assays in their own hands.

Line 444-445. Did you look for inducible clindamycin resistance by testing this strain with D-zone test?

Line 473-475. I would be very cautious using lytA as a SYBR green assay without the probe. I would be concerned about false positives from oral streptococci.

Line 490-491. In your concluding sentences you make reference to qPCR for 3 main bacteria? I don’t see any of this data in the paper and not really sure of the relevance to the data you are showing in this paper.

6. PLOS authors have the option to publish the peer review history of their article (what does this mean?). If published, this will include your full peer review and any attached files.

Reviewer #1: No

Reviewer #2: No

---

## [Author Response · Author response to Decision Letter 0]

17 May 2022

On behalf of the authors, I would like to thank you both reviewers for spending time in the correction of our manuscript and for all the comments which were important contributions to improving the manuscript quality. The text has been accordingly corrected and reviewed, and missing information was added. All changes in the text have been highlighted. 

Moreover, the manuscript was reviewed by an English native teacher, with many years of experience in professional editing of scientific articles. We believe that with the valuable aid of the reviewers the manuscript has been significantly improved, and would be now accepted for publication.

Reviewer #1:

1) The authors should clearly state the benefits of adapting the existing probe-based assay to a melting-curve based assay. If the motivation is economical, a paragraph stating the price differences should be presented and a convincing argument presented to support this. 

The economic importance of this new assay was stated in the discussion. 

2) The detailed description of the statistical tests employed should be shortened and only the relevant sections included. The main message of the paper is the method and its validation, and the paper should focus on this. The entire section on the choice of Tm prediction software (and the related sections in the discussion) should be removed. It is nonsensical to choose a prediction software and method based on its agreement with the empiric determined Tm, as is stated in lines 379-380. 

According to the statistical test applied, we observed that the differences in Tm were or not statistically significant. Therefore, we thought it would be better to explain the reason for our choices and, for example, why the same samples as CSF with two different types of extraction were kept separately in different groups. However, we agreed this is not the focus of our manuscript. Thus, statistical analyses were shortened as you suggested. 

The section about prediction software was replaced and modified. What we meant was not clear, so we hope that this modification solved the problem. We believe it is important to keep the message for readers that these software pieces produce extremely different results when compared with the empirically determined Tm, and the software applied in this study proved to be the most reliable because it came closest to the real values. This is extremely important in the design of primers to be applied in this kind of methodology (SYBR Green qPCR) to obtain a PCR fragment with size and Tm as expected.

3) The paper should clearly state what were the validation conditions used. If this was against the PCR on the DNA extracted from all the strains or, which would me much more interesting, against the DNA extracted from the CSF from which some of the strains were isolated. The discussion of results with different PCR conditions than the ones used (ex: lines 291-297) should be deleted. 

We agreed with the reviewer and removed results with different PCR conditions. The conditions under which the method was standardized were described in the “Standardization of the SYBR Green qPCR multiplex” section and they were validated with DNA from strain and for CSF samples. The sentence with the criteria for acceptance validated in this study was slightly adapted to be clearer.

4) The susceptibility of all isolates to all targeted antimicrobials must be known (table 2). 

Antibiotic susceptibility tests were performed to complete the missing data from 4 standard strains and table 2 and the text were modified accordingly. 

5) Given the variations in melting curves seen when DNA was prepared from different sources, what would be the robustness of the assay when run in different platforms? Although the Roche Lightcycler 480 II is quite a popular platform it is by no means the only one. A few sentences discussing the potential portability of the assay to other platforms would be important for a wider interest in the paper. 

The statistical difference observed for Tm of fragments obtained from different sources of DNA (strain, CSF from column, and heating) was not expected. But it does not mean the assay is not applicable on different platforms. This phenomenon will happen no matter the platform since we observed it in other studies from our laboratory. Probably, the criteria for acceptance should be adapted according to the platform. A statement about this was added in the discussion section. 

 6) The purpose of the retrospective analysis and how it could impact on the clinical management of the patients or how the information could help in the surveillance of pneumococcal infections in this area must be clarified or this section removed. The fact that macrolides and lincosamides are not used in the treatment of pneumococcal meningitis raises the question of the usefulness of determining the potential susceptibility to these antimicrobials, a point that should be addressed. 

The retrospective analysis can be useful in the surveillance of pneumococcal infections in the region, and it is shown that the methodology applied here can help to monitor the epidemiology of the pneumococcal disease, even in the absence of isolated strain. A sentence addressing this issue was included in the discussion.

7) The discussion should be shortened and focused on the main message of the paper without repetition of what was stated in the results section 

In the discussion, we intended to explain all findings in our study, besides the main message of the paper which is the determination of resistance genes in samples without isolated strains, with a low-cost methodology. The discussion was changed accordingly in the revised manuscript.

Other points 

8) Line 121. The authors changed the reverse primer amplifying the mef gene from the previously published assay. Why was this done? A sentence should be included to explain this option. 

This change was necessary to generate an amplicon with a different Tm. Using the same primers from the previous study generates a fragment of mef with the same Tm of pbp2 fragment. This explanation was added in the material and methods section.

9) English use: line 116 “and thus it is sensitivity” should be “and thus its sensitivity”; line 164 “Comparison of technics” should be “Comparison of techniques”; line 181 “Characterization of applied samples” should be “Characterization of analyzed samples”; line 291 and elsewhere “acceptation criteria” should be “acceptance criteria” 

They were changed in our revised manuscript. Also, as mentioned before, the manuscript was reviewed now by an English native teacher and experienced in scientific texts.

10) Lines 222-224. The central limit theorem is not applicable to 39 data points which is what is available for the pbp2 gene (table 1). This sentence must be removed. 

It was changed in our revised manuscript.

11) Table 1. The P values have comas instead of points separating the decimals. This table repeat the data presented in figure 1. I would suggest offering the figure as supplementary material.

It was changed in our revised manuscript. However, the information presented in figure 1 and table 1 is different. The p-value in the table presents the comparison with the Tm predicted by the software. Figure 1 presents the comparison of the empirically determined Tm from all samples of the same gene (ANOVA test).

Reviewer #2: 

Specific comments

Line 21-22 and 59. Not necessary to have capital letters for Public Health? 

We kindly request to differ from the reviewer, but according to the English teacher who revised our manuscript, "Public health" is correct, as over here “Public" works as an adjective for the noun.

Line 71. It is not clear what you mean by qPCR “provides more complete results than the standard techniques”? 

This part was removed from the text.

Line 164. “technics” should be “techniques” 

It was changed in our revised manuscript.

Line 206-208. I am curious why you would need to have 2 different primer concentrations when you use isolate vs CSF DNA? Can you explain and would you be concerned with performance of the assay with varying amounts of DNA in your sample? Did you do a serial dilution series to determine LOD? 

We believe that these two different primer concentrations were necessary due to DNA template concentration. DNA extracted from pure culture is massively more concentrated than in the CSF sample. Thus, it is necessary to have high concentrations to get a positive result when using DNA extracted from the CSF sample, whereas when these higher concentrations were applied to DNA from culture, false-positive results can be produced. In this study, we extensively tested this assay and we applied genetic material from CSF samples with different qualities, which means higher and lower bacterial DNA concentrations. Therefore, we are sure that with this higher primer concentration, it will be able to detect lower bacterial DNA concentration. Also, when using CSF samples, false-positive results will not be obtained since the amount of bacterial DNA is never equivalent to DNA from culture. We did not determine LOD because it is not possible to know the exact amount of bacterial DNA present in the CSF sample. This is a mixture of human and bacterial DNA. Therefore, measurement by optical density, for example, would be compromised. Also, serial dilution of DNA from culture would produce samples with known concentration; however, it does not correspond to the total amount of DNA present in the CSF sample, due to this mixture of DNA.

Line 267-269 and Table 1. It Is not clear why mef has only 2 groups for comparison? Why was CSF silica and CSF heating combined? Was it due to sample size?

This was previously explained in the first version of the manuscript in lines 233-235. These two groups were combined in one due to the small number of samples; three CSF samples were extracted by silica column and one of them was also extracted by heating. We could not perform the statistical analyzes using a group with only one sample. This gene was not found in abundance in the samples which were checked by the AST, only three samples. Now, the information is reinforced in the text.

 Line 292-297. Do you think that this approach using SYBR green without a probe is accurate enough given that you mention false positive and negative results with changes in temperature and primer concentrations. I did not see any data provided showing effects of sample concentration of DNA on the Tm and positive vs negative results. Can you explain in Fig 1 the large differences between Tm for CSF (silica) vs CSF (heating)? Is this related to DNA concentrations? 

We think the qPCR using SYBR green without a probe is accurate and reliable but the dissociation curve must be analyzed and Tm must be in the range established here. There is no effect of sample concentration on the positive or negative result, as observed with DNA extracted from culture is much higher than CSF sample which the same strain was isolated and they produce the same result. Surely, if you have a CSF sample with so much lower concentration, you would not have a positive result, like any other assay, due to the limit of detection. However, with certainty, from this sample would not be possible to isolate a strain due to the low bacterial concentration. What changed the outcome here was the primer concentration, and using the specified quantity of primers according to the sample (culture or CSF) should have no problems. The variation in Tm was observed with different types of extraction (silica column and heating). Unfortunately, we could not identify why this phenomenon happens. We believe this is not due to DNA concentration, since a CSF sample before the extraction with silica column and heating is the same, thus the same amount of DNA. It is hypothesized the quality of the sample could affect the Tm, for example, DNA from culture is pure, without contaminant; DNA from CSF sample purified by column has a mixture of bacterial and human DNA; DNA from CSF sample extracted by heating has many contaminants present from the fluid. 

Line 387-388. You state “Thus, an interval must be adopted to be 388 considered positive according to the origin of the genetic material”. I have my doubts on how accurate this method would be if say for example you used a pleural fluid or blood sample? Would you be confident that this assay would accurately determine penicillin, erythromycin or clindamycin susceptibility in these other specimen types give the variability of the assay? 

We believe that genetic material from different sources such as pleural fluid or blood samples would produce similar results, with Tm in the range established here. However, this was not performed in this study. As with any assay validation, further studies would be necessary to test with these kinds of samples.

Lines 399-402. Again, here you indicate that differences such as changes in PCR instrumentation would affect results? I don’t see how this method could be useful to a larger community? Perhaps validated in your own lab for testing but seems problematic in that every lab would need to revalidate the assays in their own hands. 

This sentence was misunderstood, for this reason, it was removed. Actually, changes in PCR instrumentation do not affect the results, positive or negative. What should be changed are the criteria for acceptance and the interpretation. Because each piece of equipment has itself way to present the final result and also a specific configuration.

As with any diagnostic kit, it is validated with specific equipment and reagents by the manufacturer. Commercial kits for real-time PCR can be applied in different instruments; however, the manufacturer only guarantees if applied with the condition established and validated by them. In our study, we established the condition for two types of samples (strain and CSF), with only one reagent (PowerUp™ SYBR™ Green Master Mix - Thermo Fisher Scientific) in one piece of equipment (LightCycler® 480 II - Roche). But it does not mean the assay cannot be applied on other platforms. The use of other PCR equipment is absolutely possible. However, the criteria for acceptance must be adapted. For example, the Roche system presents -d/dT fluorescence at 465-510 nm, and the Tm peak must have a height superior to one for strain and any height for CSF sample; whereas QuantStudio™ 5 Real-Time PCR System (Applied Biosystems™, Thermo Fisher Scientific) presents different fluorescence unit in the melt curve plot (derivative -Rn), and the order of magnitude is in the thousands. 

A new sentence in the discussion was included to address this issue.

Line 444-445. Did you look for inducible clindamycin resistance by testing this strain with D-zone test?

It is part of the laboratory routine of Adolfo Lutz Institute in São Paulo to evaluate the inducible clindamycin resistance by the D-zone test, but none was positive. The strains studied here showed constitutive resistance to clindamycin.

Line 473-475. I would be very cautious using lytA as a SYBR green assay without the probe. I would be concerned about false positives from oral streptococci. 

The lytA gene is the standard for PCR detection of Streptococcus pneumoniae in CSF samples. At the moment of diagnosis, these samples were tested by qPCR with a specific probe for pneumococcus detection using the lytA gene. In our study, we performed with SYBR green, only to check if the genetic material was undamaged after years in freezer storage. We removed this information from the discussion and mentioned it in the results. Also, I don’t believe that oral streptococci may be present in the CSF sample.

Line 490-491. In your concluding sentences you make reference to qPCR for 3 main bacteria? I don’t see any of this data in the paper and not really sure of the relevance to the data you are showing in this paper.

We agreed and the last sentence was modified.

---

## [Decision Letter · Decision Letter 1]

30 May 2022

Multiplex real-time PCR using SYBR Green: unspecific intercalating dye to detect antimicrobial resistance genes of Streptococcus pneumoniae in cerebrospinal fluid

PONE-D-22-02044R1

Dear Dr. Campos,

We’re pleased to inform you that your manuscript has been judged scientifically suitable for publication and will be formally accepted for publication once it meets all outstanding technical requirements.

Kind regards,

Jose Melo-Cristino, M.D., Ph.D.

Academic Editor

PLOS ONE

Additional Editor Comments (optional):

Reviewers' comments:

Reviewer's Responses to Questions

**Comments to the Author**

1. If the authors have adequately addressed your comments raised in a previous round of review and you feel that this manuscript is now acceptable for publication, you may indicate that here to bypass the “Comments to the Author” section, enter your conflict of interest statement in the “Confidential to Editor” section, and submit your "Accept" recommendation.

Reviewer #1: All comments have been addressed

2. Is the manuscript technically sound, and do the data support the conclusions?

Reviewer #1: Yes

3. Has the statistical analysis been performed appropriately and rigorously? 

Reviewer #1: Yes

4. Have the authors made all data underlying the findings in their manuscript fully available?

Reviewer #1: Yes

5. Is the manuscript presented in an intelligible fashion and written in standard English?

Reviewer #1: Yes

6. Review Comments to the Author

Reviewer #1: The authors have addressed most of the comments raised by the reviewers and provided necessary information in the text.

7. PLOS authors have the option to publish the peer review history of their article (what does this mean?). If published, this will include your full peer review and any attached files.

Reviewer #1: No

---

## [Editor Report · Acceptance letter]

6 Jun 2022

PONE-D-22-02044R1 

Multiplex real-time PCR using SYBR Green: unspecific intercalating dye to detect antimicrobial resistance genes of Streptococcus pneumoniae in cerebrospinal fluid 

Dear Dr. Campos:

I'm pleased to inform you that your manuscript has been deemed suitable for publication in PLOS ONE. Congratulations! Your manuscript is now with our production department. 

Kind regards, 

on behalf of

Prof. Jose Melo-Cristino 

Academic Editor

PLOS ONE